# A Chimeric Zika Virus between Viral Strains MR766 and BeH819015 Highlights a Role for E-glycan Loop in Antibody-mediated Virus Neutralization

**DOI:** 10.3390/vaccines7020055

**Published:** 2019-06-24

**Authors:** Etienne Frumence, Wildriss Viranaicken, Sandra Bos, Maria-Teresa Alvarez-Martinez, Marjolaine Roche, Jacques-Damien Arnaud, Gilles Gadea, Philippe Desprès

**Affiliations:** 1Université de La Réunion, INSERM U1187, CNRS UMR 9192, IRD UMR 249, Unité Mixte Processus Infectieux en Milieu Insulaire Tropical, Plateforme Technologique CYROI, 97491 Sainte-Clotilde, La Réunion, France; etienne.frumence@univ-reunion.fr (E.F.) wildriss.viranaicken@univ-reunion.fr (W.V.) sandrabos.lab@gmail.com (S.B.) marjolaine.roche@univ-reunion.fr (M.R.) gilles.gadea@inserm.fr (G.G.); 2Plateforme CECEMA, Université de Montpellier, 34095 Montpellier, France; maria-teresa.alvarez-martinez@umontpellier.fr (M.-T.A.-M.); damien.arnaud@montpellier.fr (J.-D.A.)

**Keywords:** arbovirus, Zika virus, viral clone, chimeric virus, vaccine, immunity, mouse model, humoral response, viral envelope protein, neutralizing antibody

## Abstract

Zika virus (ZIKV) is an emerging mosquito-borne flavivirus which is of major public health concern. ZIKV infection is recognized as the cause of congenital Zika disease and other neurological defects, with no specific prophylactic or therapeutic treatments. As the humoral immune response is an essential component of protective immunity, there is an urgent need for effective vaccines that confer protection against ZIKV infection. In the present study, we evaluate the immunogenicity of chimeric viral clone ZIKBeHMR-2, in which the region encoding the structural proteins of the African strain MR766 backbone was replaced with its counterpart from the epidemic strain BeH819015. Three amino-acid substitutions I152T, T156I, and H158Y were introduced in the glycan loop of the E protein (E-GL) making ZIKBeHMR-2 a non-glycosylated virus. Adult BALB/c mice inoculated intraperitoneally with ZIKBeHMR-2 developed anti-ZIKV antibodies directed against viral proteins E and NS1 and a booster dose increased antibody titers. Immunization with ZIKBeHMR-2 resulted in a rapid production of neutralizing anti-ZIKV antibodies. Antibody-mediated ZIKV neutralization was effective against viral strain MR766, whereas epidemic ZIKV strains were poorly sensitive to neutralization by anti-ZIKBeHMR-2 immune sera. From our data, we propose that the three E-GL residues at positions E-152, E-156, and E-158 greatly influence the accessibility of neutralizing antibody epitopes on ZIKV.

## 1. Introduction

Zika virus (ZIKV), belonging to the flavivirus genus of the *Flaviviridae* family, was first discovered in Africa in 1947 [1]. Phylogenetic analysis distinguished African and Asian lineages of ZIKV [2,3]. Historically, ZIKV has been known to occur in sporadic outbreaks in Africa and South-Asia. Recently, ZIKV became a public health concern with epidemics occurring in Yap islands (2007), French Polynesia (2013) and South America (2015) [4,5]. ZIKV is recognized as the cause of congenital Zika syndrome leading to severe neurodevelopmental diseases and of other neurological complications such as Guillain-Barre syndrome to adults [5,6]. Natural transmission of ZIKV in humans involves infectious mosquitoes from the *Aedes* genus, but ZIKV transmission has also been documented through sexual contact, blood transfusion and intrauterine transmission [7]. 

ZIKV contains a positive sense single-stranded RNA encoding a polyprotein which is processed co- and post-translationally by viral and host proteases to produce the three structural proteins (C, M, and E) and seven nonstructural proteins, including the NS1 protein which is secreted from infected cells as a hexameric lipoprotein particle [8,9]. The envelope E protein is involved in the virus binding to the host-cell surface and the subsequent internalization of virus particles through a receptor-mediated endocytosis [10]. In the endosomes, the E protein undergoes conformational changes in a pH-dependent manner leading to the fusion between internalized virus particles and cellular membranes and the subsequent release of genomic RNA into the cytosol. The ectodomain of ZIKV E protein is divided into three domains: a central ß-barrel shaped domain I (EDI), a finger-like domain II (EDII) and a C-terminal immunoglobulin-like domain III (EDIII) [10]. The ZIKV EDI comprises a glycan loop (EDI-GL) which may be post-translationally N-glycosylated at position E-154; EDII contains a fusion loop (EDII-FL) which contributes to virus-mediated membrane fusion; EDIII has an extended CD-loop which might play a key role in virus stability [10,11,12,13]. The E protein is the main target of neutralizing antibodies against ZIKV [14]. Their binding can prevent virus attachment to cell receptors and/or membrane fusion of internalized virus particles [10]. The neutralization of ZIKV depends on the accessibility of neutralizing antibody epitopes on the virus surface.

We recently reported the development of molecular clones MR766^MC^ and BR15^MC^ using the reverse genetic method ISA [15,16]. The construction of MR766^MC^ was based on historical African ZIKV strain MR766 isolated in a non-human primate in 1947. MR766 has undergone extensive passaging in mouse brains and cell cultures leading to virus variants that have been well documented [17]. Some variants of MR766 lack of glycosylation due to a deletion spanning the EDI-GL or mutation which causes a loss of the carbohydrate attachment site on residue Asn154. Among the deposited MR766 sequences, we decided to select the MR766-NIID variant (accession number LC002520), which contains a residue Ile at position E-156 leading to non-glycosylated E protein. The construction of BR15^MC^ was based on the deposited sequence of epidemic Brazilian ZIKV strain BeH819015 (accession number KU365778) which was isolated in Brazil in 2015 [16]. Utilizing the ISA method, we constructed a molecular clone CHIM (here referred to as ZIKBeHMR-1) derived from MR766^MC^ in which the region coding for C, prM and the E ectodomain (E-1 to E-436) from MR766 was replaced with the one of BeH819015 [16]. Genetic comparative analysis identified 16 divergent amino-acids among the E proteins of MR766^MC^ and BR15^MC^ [16]. Noteworthy, there are three amino-acid changes at positions E-152, E-156, and E-158 surrounding the residue Asn154 where glycan is linked.

The study of chimera ZIKBeHMR-1 demonstrated the implication of the structural protein region in the host-cell permissiveness of host cells to ZIKV, whereas the nonstructural protein region would be more involved in induction of innate immunity as well as cytokine production [16]. All these characteristics make ZIKBeHMR-1 an attractive viral clone that could provide anti-ZIKV immunity following immunization. Given that pathogenic properties of ZIKV relate to the N-glycosylation status of E [18,19,20,21,22], we generated a mutant derived from ZIKBeHMR-1 (here referred to as ZIKBeHMR-2) which lacks N-glycosylation. The purpose of our study was to evaluate the ability of the mutant viral clone ZIKBeHMR-2 to induce anti-ZIKV neutralizing antibodies in a mouse model. We showed that anti-ZIKBeHMR-2 immune sera neutralize ZIKV in a viral strain-dependent manner. Our data suggest that EDI-GL residues influence accessibility of the epitopes targeted by the neutralizing anti-ZIKV antibodies.

## 2. Materials and Methods 

### 2.1. Cells, Viruses, and Reagents

Vero cells (ATCC, CCL-81) were cultured at 37 °C under a 5% CO_2_ atmosphere in MEM medium supplemented with 5% heat-inactivated fetal bovine serum (FBS) and 0.002 M L-Glutamine, 0.001 M sodium pyruvate, 100 U/mL of penicillin, 0.1 mg/mL of streptomycin and 0.5 µg/mL of fungizone (PAN Biotech, Aidenbach, Germany). The ZIKV strain PF-25013-18 isolated in French Polynesia in 2013 has been previously described [23,24]. The molecular clones MR766^MC^ derived from historical African strain MR766 (Genbank access n°LC002520) and BR15^MC^ derived from epidemic Brazilian strain BeH819015 (Genbank access n° KU365778) were previously described by Gadea et al. [15] and Bos et al. [16]. ZIKV BR15 virus is available at BEI Resources (Manassas, VA) under the catalog number NR-51129 (www.beiresources.org). The chimeric viral clone ZIKV ZIKBeHMR-1 (or CHIM) which consists of BR15^MC^ structural proteins into MR766^MC^ backbone was previously described by Bos et al. [16]. The mouse anti-*pan* flavivirus envelope E protein mAb 4G2 was produced by RD Biotech (France). Rat anti-ZIKV NS1 immune serum was previously described elsewhere [9]. The horseradish peroxidase-conjugated mouse anti-His_tag_ mAb was purchased from Cell Signaling Technology. Donkey anti-mouse Alexa Fluor 488 antibodies were purchased from Invitrogen and horseradish peroxidase-conjugated anti-mouse antibodies from Abcam. Horseradish peroxidase-conjugated anti-rat antibodies were purchased from Vector Labs.

### 2.2. Recovering of Viral Clone ZIKBeHMR-2

The chimeric ZIKV clone ZIKBeHMR-1 consists of MR766^MC^, in which the structural protein region encoding the C, prM, and the N-terminal region of E (E-1 to E-436) was replaced by the counterpart of BR15^MC^. The mutant ZIKBeHMR-2 was produced by using the synthetic genes Z1^BR15 (E-I152T, E-T156I, E-H158Y)^, Z2^MR766-MC^, Z3^MR766-MC^, and Z4^MR766-MC^ cloned into plasmid pUC57 (Genecust, France). Z1^BR15-MC^ codes for the structural proteins of BeH819015 (C-1 to E-436) and contains the unique restriction site *Pvu* I as a viral clone marker. The design of mutant Z1^BR15 (E-I152T, E-T156I, E-H158Y)^ bearing the E amino-acid residues Thr52 (codon ACT), Ile156 (codon ATA) and Tyr158 (codon TAT) was based on the original sequence of Z1^BR15-MC^. The fragments Z2 to Z4 of MR766^MC^ were described elsewhere [15]. The production strategy of ZIKBeHMR-2 by the reverse genetic ISA method was made as previously described [15]. Briefly, the four viral genes were amplified by PCR from their respective plasmids and the PCR fragments were electroporated into Vero cells. Viral titers were determined by a conventional plaque-forming assay, as previously described [15,23]. Viral titers on Vero cells were expressed as plaque-forming units per mL (PFU/mL). 

### 2.3. Cytotoxicity Assay

Cell viability was determined by standard MTT assay as previously described [23]. Thiazolyl Blue Tetrazolium Bromide (MTT) (Euromedex Souffelweyersheim, France) at 5 mg/mL was added on cells cultured in a 96-well plate. Following 1 h incubation, MTT medium was removed and the insoluble formazan was solubilized with 0.1 mL of DMSO. Absorbance of converted dye was measured at 570 nm with a background subtraction at 690 nm. Viability was expressed as the percentage of cell metabolic activity in ZIKV-infected cells relative to those in mock-infected cells. 

### 2.4. Immunodetection of Viral Proteins

For immunofluorescence assay, cells were fixed with 3.7% paraformaldehyde in PBS and permeabilized with 0.15% Triton X-100 in PBS. ZIKV E protein was detected with the anti-*pan* flavivirus E mAb 4G2 (1:500), followed by incubation with AlexaFluor488-conjugated secondary antibody (1:500) and DAPI for the staining of nucleus (100 ng/mL). The slides were examined using a fluorescent Nikon Eclipse E2000-U microscope (Nikon, Tokyo, Japan). Images were obtained using a Hamamatsu ORCA-ER camera and the imaging software NIS-Element AR (Nikon). 

Western blot assay was essentially performed as previously described [23,25]. Briefly, cells were lyzed with RIPA lysis buffer (Sigma, Lyon, France) and concentration of proteins in the cell lysates was quantified using the BCA protein assay. Proteins in samples were separated on 4–12% PAGE-SDS gels and transferred to nitrocellulose membranes. The blots were reacted with mouse serum (1:500) and then horseradish peroxidase (HRP)-conjugated goat anti-mouse antibodies (1:2000). Revelation was done using Amersham ECL prime Western Blotting Detection Reagent (GE Healthcare, Chicago, IL, USA) and exposed on an Amersham imager 600 (GE Healthcare). All Western-blot data are representative of three independent experiments.

### 2.5. Mouse Experiment and Ethical Statement

Groups of 5-week-old female BALB/cJRj mice (*n* = 5) were intraperitoneally (i.p.) inoculated with various doses of live ZIKBeHMR-2 virus in 0.1 mL PBS/0.2% FBS. Mice inoculated once with 0.1 mL PBS/0.2% FBS, cultured Vero cell supernatant or inactivated ZIKBeHMR-2 (5 log PFU equivalent dose), served as negative controls. Animals were anesthetized with isoflurane before any experimental procedure. Blood samples were obtained from retro-orbital sinus. The challenged mice were monitored daily for signs of morbidity and mortality. Three independent experiments were performed, totaling 18 groups of five mice. Two groups of mice received 2 or 3 log PFU. Three groups of mice received 4 log PFU. Ten groups of mice received 5 log PFU. Three groups served as negative controls. Immunized mice received a booster inoculation with the same dose 6 weeks later, and samples were collected in week 10.

All mice were housed under pathogen-free conditions at the ECE (Etablissement Confiné d’Expérimentation) animal facility from the CECEMA (Centre d’Elevage et de Conditionnement Expérimental des Modèles Animaux; Université de Montpellier 2, France; n° B3417234). The protocols and subsequent experiments were ethically approved by the Ethic Committee for Control of Experiments on Animals at the CECEMA with the reference n° 036 and by the French Ministère de l’Enseignement Supérieur, de la Recherche et de l’Innovation with reference APAFIS#9137-2017030316134494 v6 (February 28th, 2018). Mouse experiments were conducted according to the recommendations of the SBEA from CECEMA and the PREPARE guidelines for planning animal research and testing [26] as part of an NC3Rs initiative to improve the design, analysis and reporting of research using animals.

### 2.6. Preparation of ZIKBeHMR-2 Samples for Mouse Experiments

Vero cells were infected 5 days with a low-passaged ZIKBeHMR-2 virus stock (7.7 log PFU/mL) at the multiplicity of infection (m.o.i.) of 0.1 PFU per cell. Cell supernatants were recovered, clarified and then titer of the viral stocks in MEM/2%FBS was determined. Ten-fold serial dilutions of the ZIKBeHMR-2 virus suspension were prepared in PBS/0.2%FBS to cover a range from 1/10 to 1/1000. The virus dilutions were aliquoted into individual 1-mL vials and then stored at −80 °C. A sample of each virus dilution stored at −80 °C was thawed and titrated on Vero cells for its residual infectivity. Mice received an infectious viral dose corresponding to virus titer that has been obtained after thawing of the vials stored at −80 °C. To inactivate ZIKBeHMR-2, virus samples (5 log PFU/mL) were heated at 56 °C for 30 min and the absence of remaining infectivity was verified by plaque-forming assay. Aliquots of sterile PBS/0.2%FBS (vehicle) and mock-infected Vero cell supernatants served as negative controls.

### 2.7. Indirect ELISA

Indirect ELISA measured the production of anti-ZIKV IgGs in immunized mice. UV-inactivated ZIKBeHMR-2 samples in MEM/2%FBS were coated on 96-well ELISA plates at the concentration of equivalent 5 log PFU per well in 0.1 mL PBS overnight. After washing 3 times with PBS-0.05% Tween, 0.1 mL of PBS-milk 3% was added as a blocking agent. Serum samples were two-fold serial diluted in 3% milk in PBS-tween 0.05% with a starting dilution of 1:50. Plates were incubated with mouse serum samples and then incubated in the presence of HRP-conjugated anti-mouse IgG antibody (1:2000). Plates were incubated with TMB substrate (eBioscience) and then stopped with 1N HCl. Absorbance was measured using microplate reader at 450 nm. 

To measure the production of anti-ZIKV EDIII IgGs in immunized mice, indirect ELISA was performed using recombinant Domain III of the E protein (rEDIII) from epidemic ZIKV strain PF-25013-18 (GenBank number KJ776791) as previously described [6]. The EDIII from PF-25013-18 and BeH819015 share 100% of amino-acid identity. The recombinant soluble SNAP-tagged rEDIII protein was produced in insect cells and highly purified from cell supernatant as previously described [27,28]. The purified rEDIII from dengue virus (DENV) type 2 and yellow fever virus (YFV) 17D vaccine were used as viral antigen controls. For indirect ELISA, ELISA plates were coated with 0.1 mL of rEDIII at the concentration of 1 µg/mL diluted in PBS and incubated overnight at 4 °C. ELISA test was then performed as described above. The end-point titer was calculated as the reciprocal of the last immune serum dilution eliciting twice the optical density (O.D.) of sera from mice that received PBS/FBS vehicle.

### 2.8. Seroneutralization Assay

The neutralizing ability of mouse serum antibodies against ZIKV was determined by plaque reduction neutralization tests on Vero cells. Mouse sera were heat inactivated at 56 °C for 30 min. Serum samples were two-fold serial diluted in DMEM/2%FBS with a starting dilution of 1:50, and incubated for 2 h at 37 °C with an equal volume of viral suspension containing approximatively 100 PFU of ZIKV. Remaining viral infectivity was determined on Vero cells by plaque-forming assay. The end-point titer was calculated as the reciprocal of the highest serum dilution tested that reduced the number of PFU by 50% (PRNT_50_). PRNT_50_ values were determined from a nonlinear regression analysis.

### 2.9. Statistical Analysis

Data obtained between different treatments were compared by one-way or two-way ANOVA tests, as appropriate. Values of *p* < 0.05 were considered statistically significant for a post-hoc Tukey’s test. All statistical tests were done using the software Graph-Pad Prism version 6.01. Degrees of significance are indicated in the as follow: * *p* < 0.05; ** *p* < 0.01; *** *p* < 0.001, **** *p* < 0.0001, ns = not significant.

## 3. Results

### 3.1. Characterization of Viral Clone ZIKBeHMR-2

Viral clone ZIKBeHMR-1 is a chimeric ZIKV in which the structural protein region of molecular clone MR766^MC^ was replaced with that of molecular clone BR15^MC^ derived from epidemic Brazilian strain BeH819015 [16] (Figure 1). The ZIKBeHMR-1 E protein comprises the E-ectodomain (EDI to EDIII) and a part of the E-stem region from BR15^MC^. The EDI domain of ZIKBeHMR-1 contains a N-linked glycosylation site at positions E-154 to E-156. Several reports demonstrated that the glycan at Asn154 contributes to the virulence of ZIKV [13,18,19,21,22]. Consequently, we constructed a non-glycosylated viral clone ZIKBeHMR-2 which differs from ZIKBeHMR-1 by the three substitutions I152T, T156I and H158Y in the E protein (Figure 1). The cluster of residues Thr152, Ile156, and Tyr158 have been found into the MR766 E protein that is non-glycosylated. From the three amino-acid substitutions introduced in ZIKBeHMR-1 to generate mutant ZIKBeHMR-2, it is expected that the Thr-to-Ile change at position E-156 causes the loss of a glycan linked to Asn154.

By immunoblot assay using anti-flavivirus E mAb 4G2, we showed that the E protein of ZIKBeHMR-2 migrated faster than both ZIKBeHMR-1 and BR15^MC^ E proteins (Figure 2A). ZIKBeHMR-2 and MR766 E proteins have comparable migration profile thus suggesting that ZIKBeHMR-2 encodes a viral envelope that is non-glycosylated. We visualized the expression of ZIKBeHMR-1 and ZIKBeHMR-2 E proteins within infected Vero cells by IF analysis using mAb 4G2 (Figure 2B). A similar pattern has been observed between the two viruses indicating that the loss of the N-glycosylation site did not cause major changes in the subcellular distribution of the E protein.

To compare the growth of viral clones ZIKBeHMR-1 and ZIKBeHMR-2, progeny virus productions in ZIKV-infected Vero cells were examined at 24, 48, and 72 h post-infection using a conventional plaque-forming assay (Figure 2C). We noted that ZIKBeHMR-1 and ZIKBeHMR-2 had comparable viral growths and virus progeny productions. Thus, a loss of N-linked glycosylation in the E protein did not cause obvious changes in the biological characteristics of ZIKBeHMR-2 in Vero cells. The viability of Vero cells infected with ZIKBeHMR-1 or ZIKBeHMR-2 similarly decreased at 48 h post-infection (Figure 2D). Thus, ZIKBeHMR-2 was as competent as ZIKBeHMR-1 at infecting Vero cells. Together these results showed that ZIKBeHMR-2 is a suitable chimeric ZIKV for further *in vivo* analysis, and as such, was required for the following experiments.

### 3.2. ZIKVBeHMR-2 Elicits anti-ZIKV Antibody Response in BALB/c Mice 

The ability of ZIKBeHMR-2 to induce humoral immune response was assessed in immunocompetent BALB/c mice. Groups of 5-week-old BALB/c were i.p. inoculated with 5 log PFU of ZIKBeHMR-2. To enhance the production of anti-ZIKV antibodies, immunized mice received a booster inoculation with the same dose 6 weeks later. As controls, mice were inoculated once with vehicle (PBS/FBS), cultured Vero cell supernatant or inactivated ZIKBeHMR-2 virus (5 log PFU equivalent dose). Mice were bled weekly after the first dose or four weeks after the second dose. Individual or pooled immune sera were tested for anti-ZIKV antibodies. Neither mortality nor morbidity was observed in adult BALB/c mice that received one or two doses of ZIKBeHMR-2 by the i.p. route.

To evaluate humoral immune response induced by ZIKBeHMR-2, indirect ELISA was performed on pooled mouse immune sera (*n* = 5) using inactivated ZIKBeHMR-2 virus particles as capture antigen (Figure 3A). Inoculation of a single dose of ZIKBeHMR-2 induced production of specific antibodies with a mean titer reaching 194 after one month. Primed animals that received a booster dose showed a ten-fold increase in anti-ZIKV antibody titer reaching 2180. The reactivity of antibodies raised in mice after ZIKBeHMR-2 immunization was evaluated by immunoblotting using RIPA lysates of ZIKBeHMR-2-infected Vero cells as a source of viral antigens (Figure 3B). It is worthy of note that ZIKBeHMR-2 encodes a chimeric E protein with the majority of the protein (87%) coming from BR15^MC^ (or ZIKV strain BeH819015) followed by NS1 to NS5 proteins from MR766^MC^. Cell lysates were tested with pooled serum samples at the dilution 1:500. One month after the prime, BALB/c mice that received ZIKBeHMR-2 displayed specific antibodies mostly directed against NS1 dimer, and to a lesser extent, against E. The inoculation of a booster dose increased the reactivity of antibodies against E and NS1 proteins (Figure 3B). Thus, ZIKV E and NS1 antigens are the two major viral targets for humoral immune response raised against ZIKBeHMR-2.

We noted that a single dose of 5 log PFU of ZIKBeHMR-2 was capable of inducing the production of anti-ZIKV antibodies in BALB/c mice inoculated by the i.p. route. Given that immunization with ZIKBeHMR-2 developed anti-E antibodies in mice, we evaluated the anti-EDIII antibody levels in immune sera of mice that received either one or two doses of 5 log PFU of ZIKBeHMR-2 with an interval of 6 weeks. Indirect ELISA was performed using a recombinant EDIII (rEDIII) domain from ZIKV strain BeH819015 as a capture antigen for specific anti-EDIII antibodies [6,28,29]. By immunoblot assay, we showed that immune sera of mice that received twice ZIKBeHMR-2 developed specific antibodies raised against ZIKV rEDIII without cross-reactivity with recombinant EDIII from DENV or YFV (Figure 3C). Immune sera obtained from BALB/c mice immunized with ZIKBeHMR-2 were tested for anti-ZIKV EDIII antibody by indirect ELISA (Figure 3D). Immune sera from BALB/c mice inoculated with 5 log PFU of ZIKBeHMR-2 gave anti-ZIKV EDIII IgG titers of about 1100 (Table 1). A booster dose increased anti-ZIKV EDIII IgG titers to reach up to 6200 (Table 1). A low variability was observed between individual mouse immune sera, justifying the use of pooled sera in subsequent experiments. Immunization with heat-inactivated ZIKBeHMR-2 or vehicles elicited anti-ZIKV rEDIII antibody titers that were below than the basal level of 50 (Table 1). Thus, inactivated ZIKBeHMR-2 was inefficient to induce anti-E antibody in mice. Taken together, these results showed that BALB/c mice inoculated with ZIKBeHMR-2 developed high titers of anti-E antibody in BALB/c mice. Also, the ability of ZIKBeHMR-2 to elicit anti-ZIK E antibody response required viral replication in inoculated mice.

### 3.3. ZIKBeHMR-2 Stimulates a Rapid Production of Neutralizing Anti-ZIKV Antibodies

The ZIKV neutralization potency of immune sera from mice that received 5 log PFU of ZIKBeHMR-2 was first evaluated on ZIKBeHMR-2 using a conventional PRNT on Vero cells (Figure 4). 

Neutralizing antibodies were detectable soon after ZIKBeHMR-2 inoculation (Figure 4A) and reached high titers after one month (Figure 4B). A single dose of ZIKBeHMR-2 (5 log PFU) gave a PRNT_50_ titer of about 500-600 (Table 1). As expected, there was no neutralizing antibodies in sera from mice that received heat-inactivated ZIKBeHMR-2 (Table 1). A booster dose of ZIKBeHMR-2 was capable to increase the production of neutralizing antibodies (Figure 4C). One month after the boost, the levels of neutralizing anti- ZIKBeHMR-2 antibodies reached a titer of 6000 (Table 1). Thus, inoculation with ZIKBeHMR-2 elicited high levels of specific antibodies capable of neutralizing a viral clone which bears BeH819015 E protein but with three amino-acid changes in the EDI-GL.

As shown in Table 1, a single dose of 5 log PFU of ZIKBeHMR-2 was effective at eliciting a high level of anti-ZIKV neutralizing antibodies. To determine whether a lower viral dose was capable of inducing anti-ZIKV antibody production after a single inoculation, animals were i.p. inoculated with a wide range of ZIKBeHMR-2 virus varying from 2 to 5 log PFU. The levels of anti-ZIKV EDIII antibodies and neutralizing anti-ZIKV antibodies were determined one month after virus inoculation (Figure 5). We noted that anti-ZIKV E antibody titers in mice immunized with ZIKBeHMR-2 were viral dose-dependent. A single inoculation by 100 PFU of ZIKBeHMR-2 elicited anti-ZIKV E antibodies indicating that immunization could be effective with a very weak dose of virus. 

The monitoring of the antibody responses during the first 4 weeks after the prime inoculation revealed the production of significant numbers anti-ZIKV E antibodies at the week 3, regardless of the viral dose being tested (Figure 5A). This is consistent with the general notion that IgG antibodies are readily detectable only after two weeks of immunization. As shown in Figure 4A, the induction of a protective anti-ZIKV humoral immunity was achieved soon after inoculation with ZIKBeHMR-2. By PRNT, we showed that immune sera of mice that received ZIKBeHMR-2 developed anti-ZIKV neutralizing activity one week after inoculation regardless the viral dose inoculated (Figure 5B). A such result suggests that IgM antibodies could account for the observed neutralization of ZIKV by anti-ZIKBeHMR-2 immune sera at the first week of immunization. Taken together, these results demonstrated the capacity of a low amount of ZIKBeHMR-2 virus to elicit a humoral immune response against ZIKV associated to a high level of anti-ZIKV neutralizing antibodies as early as one week after inoculation of a single viral dose.

### 3.4. Neutralizing Activity of Anti-ZIKBeHMR-2 Antibodies against Various ZIKV Strains

We assessed whether mice inoculated with a single dose of ZIKBeHMR-2 developed antibodies that neutralize parental viral clones BR15^MC^ and MR766^MC^ (Table 2). Immunization with a single dose of ZIKBeHMR-2 elicited high titers of neutralizing antibodies against MR766. Surprisingly, anti-ZIKBeHMR-2 antibodies failed to neutralize viral clone BR15^MC^ derived from ZIKV strain BeH819015. A such lack of neutralizing activity was also observed with the clinical isolate PF-25013-18 collected during the ZIKV epidemic in French Polynesia in 2013 [23,24] (Table 2). Only BALB/c mice inoculated with ZIKBeHMR-2 and that received a booster dose 6 weeks later developed a weak titer of neutralizing antibodies against BR15^MC^ and PF-25013-18.

The lack of neutralization against epidemic ZIKV strains of Asian lineage prompted us to determine whether anti-ZIKBeHMR-2 antibodies have neutralizing activity against ZIKBeHMR-1 that differ from ZIKBeHMR-2 by only three amino-acid changes at positions E-152, E-156, and E-158 (Table 2). PRNT assays failed to detect neutralizing activity response against ZIKBeHMR-1 in immune sera of BALB/c mice once inoculated with 5 log PFU of ZIKBeHMR-2. The production of neutralizing antibodies against ZIKBeHMR-1 required inoculation with a booster dose in primed animals (Table 2). However, the level of neutralizing anti-ZIKBeHMR-1 antibodies was weak as compared to ZIKBeHMR-2. 

We asked whether the weak neutralization of ZIKBeHMR-1 relates to a low capacity of anti-ZIKBeHMR-2 antibodies to recognize the E protein (Figure 6). Immunoblot analysis revealed that ZIKBeHMR-2 was capable to induce antibodies that can equally react with the E proteins from ZIKBeHMR-1 and ZIKBeHMR-2. It is therefore unlikely that a change in antigenicity of E among the two virus variants could be responsible for the lower capacity of anti-ZIKBeHMR-2 antibodies to neutralize the parental viral clone.

Taken together, these results showed that anti-ZIKBeHMR-2 antibodies poorly neutralize contemporary epidemic ZIKV strains of Asian lineage in contrast to what was observed with African historical MR766 strain. The relative resistance of parental viral clone ZIKBeHMR-1 to neutralization by ZIKBeHMR-2 immune sera suggests that glycan linked to Asn154 might have an impact on the accessibility of neutralizing antibody epitopes in mature virus particles. We cannot rule out the possibility that EDI-GL residues E-152, E-156, and E-158 influence antigenicity and/or immunogenicity of ZIKV E protein.

## 4. Discussion

Zika disease became a global public health priority due to congenital Zika syndrome and other neurological disorders in humans. There is an urgent need for effective vaccines which confer protection from ZIKV infection. Significant effort has been made to propose vaccine candidates against ZIKV; some of them are currently in preclinical testing and clinical trials [30,31,32]. 

Live-attenuated virus (LAV) vaccines are among the promising approaches to combatting Zika disease. It has been reported that a LAV vaccine bearing a deletion in the 3’ untranslated region of ZIKV genomic RNA provided protective immunity against ZIKV [33]. Chimeric dengue and Japanese encephalitis viruses in which their envelope proteins were replaced with the counterparts from ZIKV were capable of inducing protective immunity against ZIKV [34,35]. The production of a neutralizing anti-E antibody response plays a key role in protection against ZIKV infection [36,37,38]. Consistent with this finding, it has been reported that expression of a recombinant E protein was efficient at inducing neutralizing antibodies and protection in vivo [39,40,41,42,43]. 

The purpose of our study was to evaluate the immunogenicity of a chimeric ZIKV in which the C, prM and E genes of historical African MR766-NIID backbone were replaced with those of epidemic Brazilian strain BeH819015. The parental chimeric ZIKBeHMR-1 contains a N-linked glycosylation sequence NDT at positions E-153 to E-156. It has been shown that non-glycosylated ZIKV strains are attenuated in their ability to cause mortality and morbidity in mice [18]. Consequently, we generated the ZIKBeHMR-2 mutant which bears the amino-acid substitutions E-I152T, E-T156I, and E-H158Y making the chimeric ZIKV a non-glycosylated virus at position Asn154. It is of interest that the residues E-152T, E-156I and E-158Y are detected in MR766. Analysis of viral growth and virus-mediated cell death showed that ZIKBeHMR-1 and ZIKBeHMR-2 were similar in their replication efficacy in Vero cells. Thus, the amino-acid changes E-I152T, E-T156I, and E-H158Y had no major effect on the biological properties of chimeric ZIKV.

While immunocompromised mice were required for the assessment of the protective efficacy of anti-ZIKV vaccines [44], immunocompetent inbred mice are mostly used to evaluate immunogenicity of vaccine candidates despite their resistance to ZIKV infection. Here, adult female BALB/c were selected to evaluate the immunogenicity of chimeric viral clone ZIKBeHMR-2. As it has been previously observed with the non-glycosylated ZIKV strain MR766 [44], infection of adult BALB/c mice with the higher dose of 5 log PFU of ZIKBeHMR-2 via a peripheral route did not cause mortality nor morbidity. Immunization with a single dose of ZIKBeHMR-2 resulted in production of anti-ZIKV antibodies and a booster dose 6 weeks after the prime resulted in an increase in anti-ZIKV antibody titers. The reactivity of anti-ZIKBeHMR-2 antibodies was documented by immunoblot assay and ELISA. Mice immunized with ZIKBeHMR-2 developed both anti-E and anti-NS1 antibodies. This is consistent with the finding that E and NS1 proteins acts as the main targets for imparting humoral immunity against ZIKV. The fact that ZIKBeHMR-2 antisera contain anti-NS1 antibodies whereas inactivated virus has been found incompetent in eliciting humoral immune response against ZIKV demonstrate that ZIKBeHMR-2 replication is effective in BALB/c mice inoculated via a peripheral route. 

Antigenic domain EDIII contains flavivirus type-specific epitopes that are recognized by neutralizing anti-ZIKV antibodies [6,28,40,45]. We noted that immunization with ZIKBeHMR-2 generated high levels of anti-ZIKV antibodies directed against the EDIII from epidemic ZIKV strains of Asian lineage. A dose-response experiment showed that 2 log PFU of ZIKBeHMR-2 were sufficient to stimulate the production of anti-ZIKV EDIII antibodies in mice. A significant titer of anti-EDIII IgGs was observed two weeks after inoculation with a single dose of 4 log PFU of ZIKBeHMR-2. We can conclude that viral clone ZIKBeHMR-2 has ability to initiate humoral immune response against ZIKV.

Inoculation by a single dose of ZIKBeHMR-2 (5 log PFU) in BALB/c mice was highly efficient at eliciting high titers of neutralizing anti-ZIKV antibodies. The production of neutralizing antibodies was observed one month after the prime and a boost resulted in a 10-fold increase in neutralization titer. Interestingly, the production of neutralizing anti-ZIKV antibodies was achieved one week after inoculation by a single dose of 2 log PFU of ZIKBeHMR-2. It is likely that IgM antibodies account for the observed neutralization of ZIKV *in vitro* suggesting a role for this isotype in controlling the early phases of ZIKV infection *in vivo*. Immune sera obtained from BALB/c inoculated with ZIKBeHMR-2 were estimated for their potency to neutralize ZIKV strains of African and Asian lineages. Although the chimeric viral clone ZIKV is able to elicit neutralizing antibodies against both BR15^MC^ (a viral clone of ZIKV strain BeH819015) and clinical isolate PF-25013-18 of Asian lineage, we noted that the two viral strains were mostly resistant to neutralization by immune sera from mice inoculated with a single dose of ZIKBeHMR-2. A boost 6 weeks after the prime stimulated the production of neutralizing antibodies against these related two epidemic strains of ZIKV but their titers were 10- to 20-fold lower as compared to MR766 of African lineage. Such results prompted us to determine whether anti- ZIKBeHMR-2 antibodies neutralize the parental viral clone ZIKBeHMR-1 which is different from its mutant by only the three identified E amino-acid substitutions. As observed with BR15^MC^ and PF-25013-18, ZIKBeHMR-1 was poorly sensitive to neutralization by anti-ZIKBeHMR-2 immune sera. However, immunoblot assay showed a comparable recognition of the E proteins from ZIKBeHMR-1 and ZIKBeHMR-2 by the anti-ZIKBeHMR-2 immune sera. It is therefore unlikely that the poor neutralization sensitivity of ZIKBeHMR-1 was due to major changes in antigenic structure of the E protein. The EDI-GL could contribute to the E dimer stabilization, and thereby modulate the accessibility of the fusion loop EDII-FL of ZIKV [22]. It is of great interest to determine whether the EDI-GL residues influence accessibility of neutralizing antibody epitopes present in EDII-FL [46].

## 5. Conclusions

The major finding of this study was that molecular clone BR15^MC^ derived from epidemic Brazilian strain of BeH819015 as well as ZIKV strain PF-205013-18 isolated during the epidemic in French Polynesia in 2013 were mostly resistant to neutralization by the mouse anti-ZIKBeHMR-2 immune sera, despite the fact that both prM and E proteins of ZIKBeHMR-2 came from BeH819015. Their low sensitivity to antibody-mediated ZIKV neutralization was also observed with the parental viral clone ZIKBeHMR-1 that differs from ZIKBeHMR-2 by only the three amino-acid substitutions E-I152T, E-T156I, and E-H158Y in the glycan-loop EDI-GL. The introduction of the Ile-to-Thr change at position E-156 leads to a lack of E glycosylation in ZIKBeHMR-2 virus. This would suggest that the residue at position E-156 may play a pivotal role on accessibility of neutralizing antibody epitopes on mature virus particles. Recently, Goo et al. emphasized that EDI-GL residues surrounding the residue Asn154 would have an effect on ZIKV antigenicity regardless of the presence of a glycan [22]. On the other hand, it has been observed that cross-reactive anti-DENV antibodies can efficiently neutralize non-glycosylated ZIKV strains [46]. From our data, we propose that residues at positions E-152 and E-158 might play a key role in the weak ZIKBeHMR-2 antibody-mediated neutralization of contemporary epidemic ZIKV strains of Asian lineage. It has therefore urgent to understand the exact role of each of the residues at positions E-152, E-156, and E-158 in the accessibility of neutralizing antibody epitopes to ZIKV particles. Such a study would broaden our current knowledge of viral antigenic properties of ZIKV by providing new information that will serve in the development of effective vaccines against emerging viral strains.

## 6. Patent

The viral clone ZIKBeHMR-2 has been described in the patent intitled “Vaccine compositions comprising an attenuated mutant zika virus” under the number WO2017220748A1 (priority date 2016-06-23).

## Figures and Tables

**Figure 1 vaccines-07-00055-f001:**
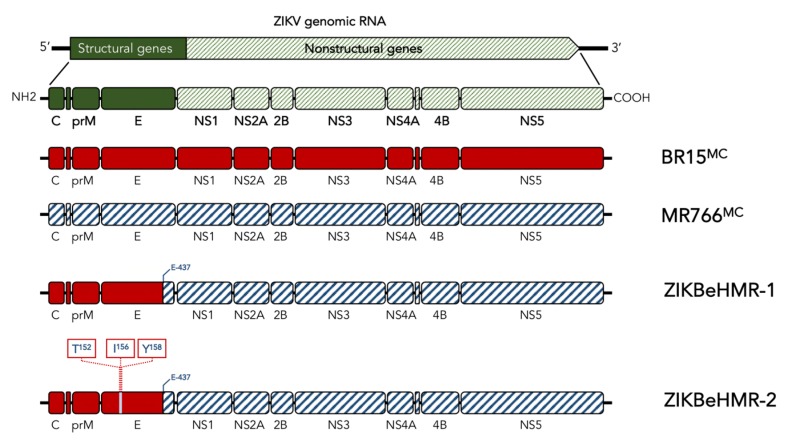
Schematic diagram of Zika virus clone ZIKBeHMR-2. The viral clone ZIKBeHMR-2 is a non-glycosylated mutant of chimeric ZIKV clone ZIKBeHMR-1, which has been described elsewhere [16]. The ZIKV structural and nonstructural genes are indicated. Chimeric ZIKBeHMR-1 genome contains the viral genomic region coding for C, prM and the N-terminal fragment (E-1 to E-436) of E from molecular clone BR15^MC^ into the MR766^MC^ backbone. The mutant ZIKBeHMR-2 bears the amino-acid substitutions I152T, T156I and H158Y. The residues Thr-152, Ile-156 and Tyr-148 have been identified in MR766 E protein.

**Figure 2 vaccines-07-00055-f002:**
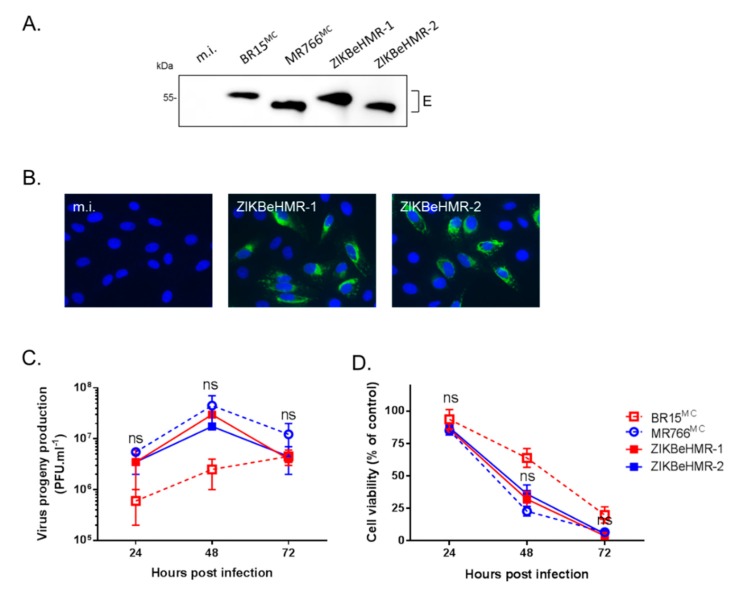
Characterization of ZIKBeHMR-2 replication in Vero cells. Vero cells were infected 24 h with viral clones BR15^MC^, MR766^MC^, ZIKBeHMR-1 or ZIKBeHMR-2 at m.o.i. of 1. or mock-infected (m.i.). In (**A**), immunoblot assay on RIPA cell lysates using anti-E mAb 4G2. In (**B**), immunofluorescence assay using mAb 4G2. Magnification: 200×. In (**C**), virus progeny production was determined by plaque-forming assay on Vero cells. Virus titers are presented as the mean (± SEM) from three independent experiments. In (**D**), the loss of viability of ZIKV-infected cells was determined using a MTT assay. Viability was expressed as the percentage of cell metabolic activity in ZIKV-infected cells relative to those in mock-infected cells. Data are presented as mean ± SEM of three independent experiments. Statistical significance between ZIKBeHMR-1 and ZIKBeHMR2 values is indicated.

**Figure 3 vaccines-07-00055-f003:**
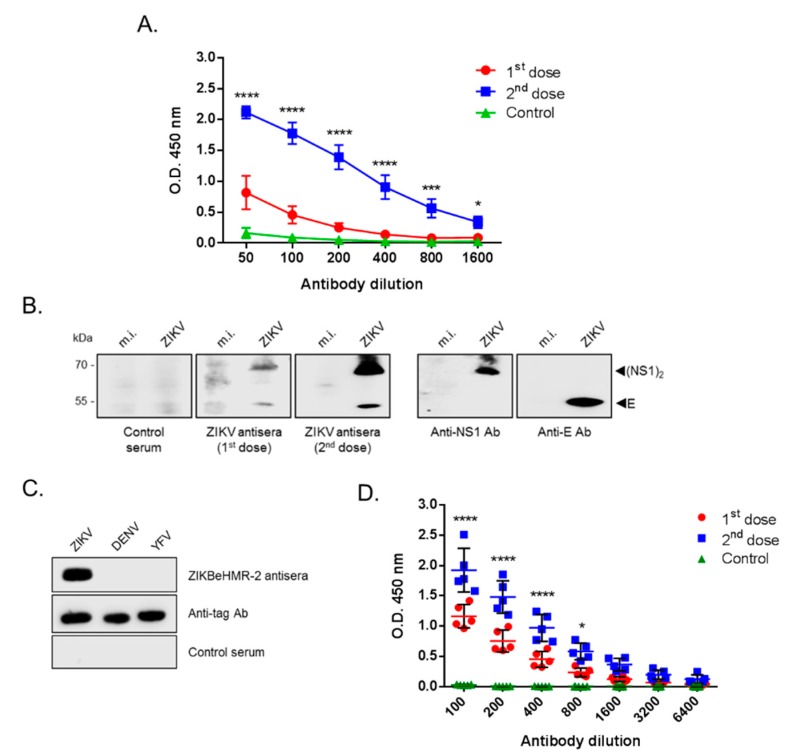
Reactivity of mouse anti-ZIKBeHMR-2 immune sera on ZIKV antigens. In (**A**), indirect ELISA on pooled immune sera (*n* = 5) of adult BALB/c mice that received one (1st dose) or two (2nd dose) doses of ZIKBeHMR-2 (5 log PFU) with an interval of 6 weeks. Inactivated ZIKBeHMR-2 virus particles served as capture viral antigen. Data are presented as mean ± SEM of three independent experiments. Statistical significance between 1st dose and 2nd dose values is indicated. In (**B**), Vero cells were infected during 24 h with ZIKBeHMR-2 (ZIKV) at m.o.i. of 1 or mock-infected (m.i.) and then lysed with RIPA lysis buffer. Immunoblot assay on cell lysates was performed with pooled immune sera of mice (ZIKV antisera) that received one or two doses of 5 log PFU of ZIKBeHMR-2 with an interval of 6 weeks. Sera of mice inoculated with vehicle were used as a control (control serum). The expression of E was verified with anti-E mAb 4G2 (anti-E Ab). The rat anti-ZIKV NS1 immune serum (anti-NS1 Ab) detected the NS1 dimer [(NS1)_2_]. In (**C**), equal amount (500 ng) of highly purified rEDIII proteins from ZIKV, DENV and YFV was analyzed by immunoblotting with pooled anti-ZIKBeHMR-2 immune sera (ZIKBeHMR-2 antisera) at dilution 1:500. Pooled sera of mice that received vehicle PBS/FBS (control serum) were used as control. Expression of His-tagged rEDIII proteins was detected using anti-HIS_tag_ mAb (anti-tag Ab). In (**D**), individual immune serum of BALB/c mice (*n* = 5) that received one (1st dose) or two doses (2nd dose) of 5 log PFU of ZIKBeHMR-2 with an interval of 6 weeks or vehicle PBS/FBS (control) were assayed for the detection of antibodies against ZIKV EDIII protein by indirect ELISA using recombinant ZIKV.rEDIII as capture antigen. Statistical significance between the first dose and second dose values is indicated.

**Figure 4 vaccines-07-00055-f004:**
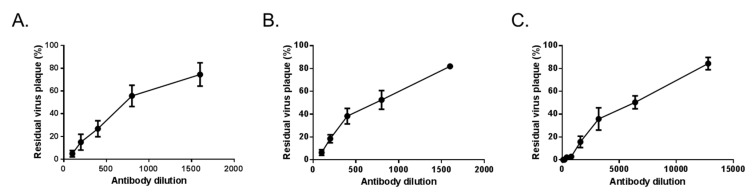
Neutralizing activity of anti-ZIKBeHMR-2 immune sera. Antibodies that neutralize ZIKBeHMR-2 were detected using a conventional PRNT. Plaque assay was performed on Vero cells. The residual virus plaques were defined as a percentage relative to the number of plaques in absence of serum. In (**A**), pooled immune sera (*n* = 5) were collected at 1-week post-inoculation with a single dose of ZIKBeHMR-2 (5 log PFU). In (**B**), immune sera (*n* = 5) were collected at 4-week post-inoculation with a single dose of ZIKBeHMR-2 (5 log PFU). In (**C**), immune sera (*n* = 5) were collected at 10-week post-inoculation with two doses of ZIKBeHMR-2 (5 log PFU) with an interval of 6 weeks. Data are expressed as the mean (± SEM) from three independent experiments.

**Figure 5 vaccines-07-00055-f005:**
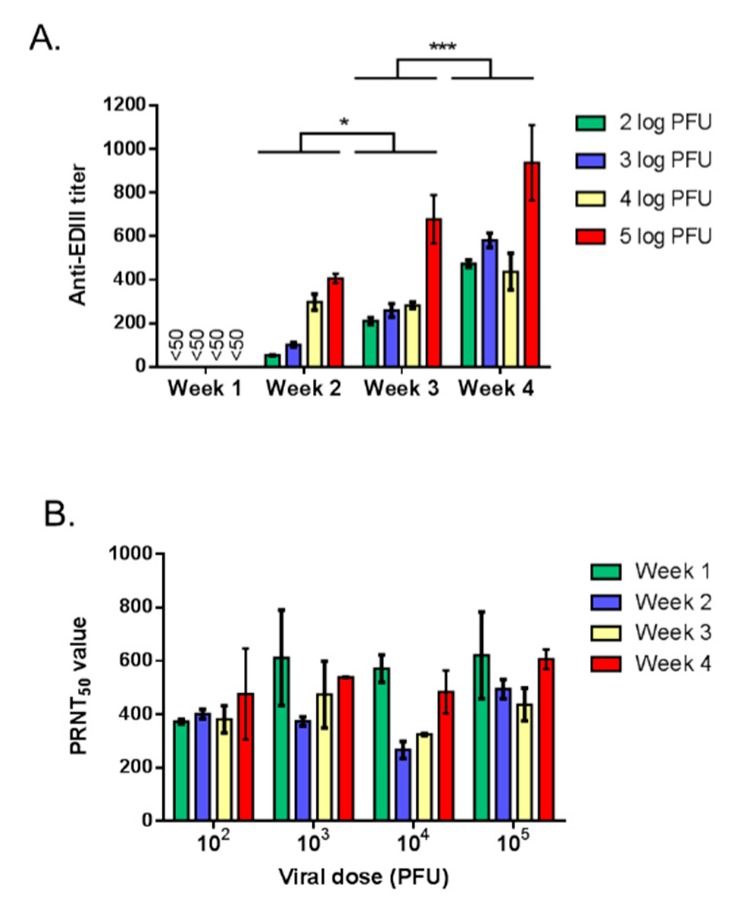
Anti-ZIKV antibody responses in mice inoculated with ZIKBeHMR-2. Groups of BALB/c mice (*n* = 5) were inoculated with increasing doses of ZIKBeHMR-2 virus (log PFU). After priming, serum samples were collected weekly (week). In (**A**), pooled immune sera were tested for the presence of IgGs directed against the EDIII by indirect ELISA using ZIKV rEDIII as capture antigen. Anti-EDIII IgG titer values are presented as the mean (±SD) from at least two independent experiments. Statistical values are indicated. In (**B**), the pooled immune sera were tested for neutralizing activity against ZIKBeHMR-2 by PRNT_50_. The values are presented as the mean (±SD) from at least two independent experiments.

**Figure 6 vaccines-07-00055-f006:**
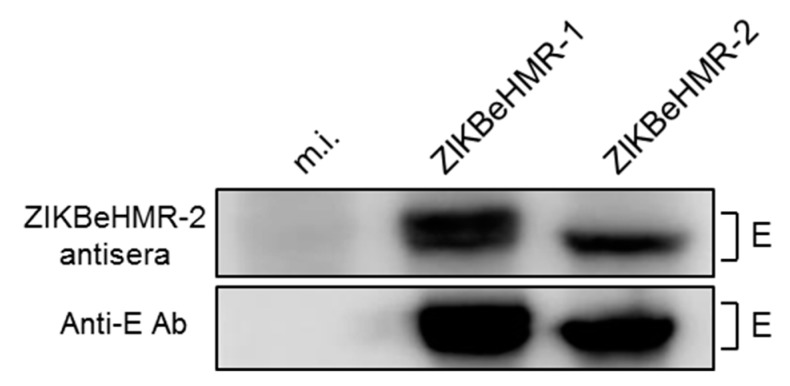
Anti-ZIKBeHMR-2 antibodies recognize the E protein from ZIKBeHMR-1. Vero cells were infected 24 h with ZIKBeHMR-1 or ZIKBeHMR-1 at m.o.i. of 1 or mock-infected (m.i.) and then lysed with RIPA lysis buffer. Immunoblot assay on cell lysates was performed with pooled immune sera of mice that received two doses of 5 log PFU of ZIKBeHMR-2 (ZIKBeHMR-2 antisera) with an interval of 6 weeks. The expression of E was verified with anti-E mAb 4G2 (anti-E Ab).

**Table 1 vaccines-07-00055-t001:** Anti-ZIKV antibody response in BALB/c mice inoculated with ZIKBeHMR-2.

Immune Sera			
**Controls ^1^**	**Weeks**	**Anti-ZIKV.rEDIII ^2^**	**PRNT_50_^3^**
PBS/FBS	4	<50	<50
Cell supernatant	4	<50	<50
Inactivated ZIKV	4	<50	<50
**ZIKBeHMR-2 ^4^**			
1st dose	4	1083 (844–1391)	529 (385–726)
2nd dose	10	6174 (3733–10212)	6032 (4039–9009)

^1^ Pooled sera from a group of BALB/c mice (*n* = 5) inoculated with one dose of vehicle PBS/FBS (PBS/FBS) or Vero cell supernatant (Cell supernatant). Pooled sera from a group of BALB/c (*n* = 5) inoculated with one dose of heat-inactivated ZIKBeHMR-2 virus (equivalent dose of 5 log PFU) (inactivated ZIKV). Serum samples were collected 4 weeks post-inoculation. ^2^ The anti-ZIKV E antibody titers in pooled immune sera were determined by indirect ELISA using ZIKV rEDIII as capture antigen. The geometrical means with 95% Confidence Interval (C.I.) of anti-EDIII IgG titer values from three independent experiments are given. ^3^Neutralizing activity of pooled immune sera using a conventional PRNT on Vero cells. PRNT_50_: the highest serum dilution that reduced the number of PFU of ZIKBeHMR-2 by at least 50%. The geometrical means with 95% Confidence Interval (C.I.) of PRNT_50_ values from three independent experiments are given. ^4^Immune sera were collected from groups of adults BALB/c (*n* = 5) that received one or two doses of 5 log PFU of ZIKBeHMR-2 with an interval of 6 weeks. Serum samples were collected 4 (prime) or 10 weeks (prime and boost) post-inoculation.

**Table 2 vaccines-07-00055-t002:** Neutralizing activity of anti- ZIKBeHMR-2 antibody against various ZIKV strains.

Viral Strain	PRNT_50_ ^1^	
	1st Dose	2nd Dose
ZIKBeHMR-1	≤50	183 (73–457)
ZIKBeHMR-2	465 (274–789)	5824 (4320–7851)
BR15^MC^	≤50	296 (159–550)
MR766^MC^	245 (124–483)	3480 (1472–8229)
PF-25013-18	≤50	170 (97–296)

^1.^ The neutralizing activity of pooled immune sera from mice that received one or two doses of 5 log PFU of ZIKBeHMR-2 with an interval of 6 weeks was evaluated against various ZIKV strains by PRNT. The geometrical means with 95% CI of PRNT_50_ values from three independent experiments are given.

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
