# Peer review of "A Chimeric Zika Virus between Viral Strains MR766 and BeH819015 Highlights a Role for E-glycan Loop in Antibody-mediated Virus Neutralization"

_vaccines, 2019, doi:10.3390/vaccines7020055_

Round 1

Reviewer 1 Report

Major 

Figure 4: the better method to determine whether "The reactivity of antibodies against E and NS1 proteins was increased in mice twice inoculated with ZIKBeHMR-2 with an interval of 6 weeks" is to perform ELISA, not WB.

Figures 4, 5 and 6 can be combined together.

Minor: 

1. typo: "the flavivirus genius of Flaviviridae family", correct genius to "genus"

2. "seven nonstructural proteins NS1 to NS5" is misleading, remove "NS1 to NS5".

Author Response

                                                                                                         June 15nd 2019

Ref: vaccines-525906

Dear Editor,

Thank you for your high consideration of our submitted manuscript # vaccines-525906.

Two major changes made to the article can be found below.

#1. We have modified the title of manuscript in the revised version according to the reviewer #2’s request.

#2. In the revised version of the manuscript, we provided a new Figure 3A. Indirect ELISA was performed on mouse immune sera from mice that received one or two doses of chimeric Zika virus clone ZIKBeHMR-2 using inactivated ZIKV as viral antigen. We showed that a boost of primed mice significantly increased anti-ZIKV antibody level.

Responses to Reviewers 1-3:

We thank the three Reviewers for their thoughtful comments and the opportunity to revise our manuscript. We believe that we have been able to address the majority of their concerns and that this has improved the manuscript significantly. Please find below a list of the Reviewers’ comments, each of which is followed by our response. 

# Reviewer 1:

Figure 4: the better method to determine whether "The reactivity of antibodies against E and NS1 proteins was increased in mice twice inoculated with ZIKBeHMR-2 with an interval of 6 weeks" is to perform ELISA, not WB.

We appreciate this point raised by the reviewer #1.  We have generated a new Figure 4B in the revised manuscript. By indirect ELISA using inactivated ZIKBeHMR2 as capture viral antigen, we showed that inoculation of two viral doses with an interval of 6 weeks gave higher anti-ZIKV antibody titers than a single dose. The text was modified in the revised manuscript accordingly.

Figures 4, 5 and 6 can be combined together.

In the revised manuscript, a new Figure 3 has been created enclosing the previous Figures 4- 6. The text was modified in the revised manuscript accordingly.

Minor: 

 typo: "the flavivirus genius of Flaviviridae family", correct genius to "genus".

Please accept our apologies for this typographical error. It has been corrected accordingly in the revised version of the manuscript

 "seven nonstructural proteins NS1 to NS5" is misleading, remove "NS1 to NS5".

Please accept our apologies for this error. The end of sentence was modified in the revised manuscript accordingly.

Sincerely yours,

Philippe DESPRES, PhD                                             

La Reunion island University                           

& UM 134 PIMIT                                                       

CYROI, 97491 Sainte-Clotilde, France            

[email protected]

Reviewer 2 Report

Overall, the experimental design are systematically performed.

The authors suggest that the three glycan loop residues E-152, E-156, and E-158 might play an important role in the accessibility of neutralizing antibody epitopes on ZIKV.

Suggestions for improvement follow.

Minor comments

1.      The words in the title “clone ZIKBeHMR-2” should be removed. A suggestion for the title could be “Chimeric Zika virus between African and Brazil strain highlights a role for E-glycan loop in antibody-mediated Zika virus neutralization”

2.      Chimeric viruses are useful to determine a role for some viral genes. Could the authors discuss why did not just introduce the mutations in the BR15MC? Was it because ZVMR766 strain induce a better humoral responses?

3.       Figures 1, 2 and 3 should be regrouped in an only Figure.

4.       Figures 4, 5 and 6 should be regrouped in an only Figure.

5.      The authors should indicate statistical significance for data presented in the Figures 3, 6 and 7.

6.      Line 335: change the words “anti-ZIKV.E” by anti-ZIKV E.

7.      Potent Zika–dengue virus cross-neutralizing antibody have been characterized previously and showed that targets a quaternary epitope that bridges two Env protein subunits (PMID: 27338953). So, they showed that cross-neutralizing antibody interact with the viral glycan on E protein. Authors should discuss this paper to highlights a role for E-glycan loop in antibody-mediated Zika virus neutralization.

Author Response

                                                                                                         June 15nd 2019

Ref: vaccines-525906

Dear Editor,

Thank you for your high consideration of our submitted manuscript # vaccines-525906.

Two major changes made to the article can be found below.

#1. We have modified the title of manuscript in the revised version according to the reviewer #2’s request.

#2. In the revised version of the manuscript, we provided a new Figure 3A. Indirect ELISA was performed on mouse immune sera from mice that received one or two doses of chimeric Zika virus clone ZIKBeHMR-2 using inactivated ZIKV as viral antigen. We showed that a boost of primed mice significantly increased anti-ZIKV antibody level.

Responses to Reviewers 1-3:

We thank the three Reviewers for their thoughtful comments and the opportunity to revise our manuscript. We believe that we have been able to address the majority of their concerns and that this has improved the manuscript significantly. Please find below a list of the Reviewers’ comments, each of which is followed by our response. 

Minor comments

#. The words in the title “clone ZIKBeHMR-2” should be removed. A suggestion for the title could beChimeric Zika virus between African and Brazil strain highlights a role for E-glycan loop in antibody-mediated Zika virus neutralization”.

We appreciate this point raised by the reviewer #2.  A new title has been inserted into the revised manuscript accordingly « A chimeric Zika virus between viral strains MR766 and BeH810915 highlights a role for E-glycan loop in antibody-mediated virus neutralization ».

#. Chimeric viruses are useful to determine a role for some viral genes. Could the authors discuss why did not just introduce the mutations in the BR15MC? Was it because ZVMR766 strain induce a better humoral responses ?

We appreciate this point raised by the reviewer #2.  As it has been stated in the manuscript lines 77 to 81 in the original version of the manuscript, viral clone ZIKBeHMR-1 (or recombinant CHIM as it had been mentioned in publication by Bos et al., Virology 516 :265, 2018) showed in a greater propensity to grow in human host cells when compared to epidemic Brazilian strain BeH819015. We rely on the idea that MR766 backbone into ZIKBeHMR-1 would confer to chimeric viral clone a greater capacity to replicate in vivo whereas production of neutralizing specific antibodies against epidemic ZIKV strains relates to expression of BeH819015 envelope proteins.

#. Figures 1, 2 and 3 should be regrouped in an only Figure.

The Figures 2 and 3 have been fused into a new Figure 2 in the revised version of the manuscript accordingly. We have modified the graph of Figure 1 which has been kept as individualized one.

#. Figures 4, 5 and 6 should be regrouped in an only Figure.

We appreciate this point raised by the reviewer #2.  The Figures 4-6 have been fused into a new Figure 4 in the revised version of the manuscript accordingly. The text was modified in the revised manuscript accordingly.

#. The authors should indicate statistical significance for data presented in the Figures 3, 6 and 7.

We agree with the reviewer #2. We have made the changes in the graph legends of new Figures 2C-D, 3D, and 4A. The text was modified in the revised manuscript accordingly.

#. Line 335: change the words “anti-ZIKV.E” by anti-ZIKV E.

Please accept our apologies for this typographical error. This was modified in the revised manuscript accordingly.

#. Potent Zika–dengue virus cross-neutralizing antibody have been characterized previously and showed that targets a quaternary epitope that bridges two Env protein subunits (PMID: 27338953). So, they showed that cross-neutralizing antibody interact with the viral glycan on E protein. Authors should discuss this paper to highlights a role for E-glycan loop in antibody-mediated Zika virus neutralization.

We greatly appreciate this point raised by the reviewer #2. Consequently, we have included the article written by Barba-Spaeth et al., (Nature 536 :48, 2016) in the References section. The main article data were discussed in Discussion section of the revised manuscript accordingly.

Sincerely yours,

Philippe DESPRES, PhD                                             

La Reunion island University                           

& UM 134 PIMIT                                                       

CYROI, 97491 Sainte-Clotilde, France            

[email protected]

Reviewer 3 Report

Etienne et al produced chimeric Zika virus constructs, vaccinated mice, and analyzed antibody production and neutralization. Vaccination with a virus lacking a glycosylated E protein produced antibody against NS1 and E, although the antibodies only effectively neutralized virus that also lacked the glycosylation site in E.

Antibody recognition of denatured protein on an immunoblot is very different from antibody binding to intact virions. The rationale involved with Fig 8 (antibodies can still react to ZIKVBeMR-1 E protein) is flawed (lines 366-369). Can you look at the levels of antibody able to interact with intact viral particles (ELISA or pull-down). Removing a glycan will open up additional epitopes that are simply unavailable when the sugar is present.  Comparing responses between ZIKBeHMR-1 and ZIKBeHMR-2 would have helped.

Minor suggestions:

Line 35: suggest changing “identified” to “distinguished”

Line 40: suggest changing “in” to “to”

Line 41: suggest changing “infected” to “infectious” and remove “as occurring”

Figure 1: When printed in black and white, one can note distinguish the color changes in the chimeric viruses. If you alter the tone of either the red or blue it would be easier to see.

Line 259: change to “ZIKBeHMR-2 encodes a chimeric E protein, with the majority of the protein (87%) coming from BeH819015 . . .”

Line 264: The immunoblots really only show the sizes for E and NS1.  Were any other proteins detected on the blot?

Figure 6: Altering color tone and/or shape of markers would help analyze the data when printed in black and white.

Line 324: While ZIKBeHMR-2 contains BeH819015 C and prM, there is a mutation in the E protein.

Line 388: or the presence of a glycan in E prevents neutralization when the immunogen lacked the glycan.

Author Response

                                                                                                         June 15nd 2019

Ref: vaccines-525906

Dear Editor,

Thank you for your high consideration of our submitted manuscript # vaccines-525906.

Two major changes made to the article can be found below.

#1. We have modified the title of manuscript in the revised version according to the reviewer #2’s request.

#2. In the revised version of the manuscript, we provided a new Figure 3A. Indirect ELISA was performed on mouse immune sera from mice that received one or two doses of chimeric Zika virus clone ZIKBeHMR-2 using inactivated ZIKV as viral antigen. We showed that a boost of primed mice significantly increased anti-ZIKV antibody level.

Responses to Reviewers 1-3:

We thank the three Reviewers for their thoughtful comments and the opportunity to revise our manuscript. We believe that we have been able to address the majority of their concerns and that this has improved the manuscript significantly. Please find below a list of the Reviewers’ comments, each of which is followed by our response. 

#. […]. The rationale involved with Fig 8 (antibodies can still react to ZIKVBeMR-1 E protein) is flawed (lines 366-369). Can you look at the levels of antibody able to interact with intact viral particles (ELISA or pull-down). Removing a glycan will open up additional epitopes that are simply unavailable when the sugar is present. 

We appreciate this point raised by the reviewer #3. We agree that the suggested experiment would provide valuable information regarding the interaction of antibody with virus particle. Experiments are currently undertaken to investigate the molecular interaction of anti-ZIKBeHMR-2 antibodies with E protein from MR766 or BR15. We would be pleased to submit our data in a next manuscript.

#. […]. Comparing responses between ZIKBeHMR-1 and ZIKBeHMR-2 would have helped.

So far animal experiments with ZIKBeHMR-1 have not been planned and we apologize for not being able to provide information on the ability of ZIKBeHMR-1 to elicit humoral immune response in this current manuscript.

Minor suggestions:

#. Line 35: suggest changing “identified” to “distinguished”

The change has been made in the revised version of the manuscript accordingly.

#. Line 40: suggest changing “in” to “to” ;

The change has been made in the revised version of the manuscript accordingly.

#. Line 41: suggest changing “infected” to “infectious” and remove “as occurring”

The change has been made in the revised version of the manuscript accordingly.

#. Figure 1: When printed in black and white, one can note distinguish the color changes in the chimeric viruses. If you alter the tone of either the red or blue it would be easier to see.

The change has been made in the revised version of the manuscript accordingly.

#. Line 259: change to “ZIKBeHMR-2 encodes a chimeric E protein, with the majority of the protein (87%) coming from BeH819015 . . .”

 The change has been made in the revised version of the manuscript accordingly.

#. Line 264: The immunoblots really only show the sizes for E and NS1.  Were any other proteins detected on the blot ?

We appreciate this point raised by the reviewer #3. As it is stated in the main text, in our hands, only intracellular forms of the E and NS1 proteins were clearly detected with anti-ZIKBeHMR-2 immune sera by immunoblot assay.

#. Figure 6: Altering color tone and/or shape of markers would help analyze the data when printed in black and white.

The change has been made in the revised version of the manuscript accordingly.

#. Line 324: While ZIKBeHMR-2 contains BeH819015 C and prM, there is a mutation in the E protein.

The change has been made in the revised version of the manuscript accordingly.

#. Line 388: or the presence of a glycan in E prevents neutralization when the immunogen lacked the glycan.

The change has been made in the revised version of the manuscript accordingly.

Sincerely yours,

Philippe DESPRES, PhD                                             

La Reunion island University                           

& UM 134 PIMIT                                                       

CYROI, 97491 Sainte-Clotilde, France            

[email protected]